# On the Fate of Butyl Methoxydibenzoylmethane (Avobenzone) in Coral Tissue and Its Effect on Coral Metabolome

**DOI:** 10.3390/metabo13040533

**Published:** 2023-04-07

**Authors:** Fanny Clergeaud, Maeva Giraudo, Alice M. S. Rodrigues, Evane Thorel, Philippe Lebaron, Didier Stien

**Affiliations:** 1Sorbonne Université, CNRS, Laboratoire de Biodiversité et Biotechnologies Microbiennes, UAR3579, Observatoire Océanologique, 66650 Banyuls-sur-Mer, France; fanny.clergeaud@obs-banyuls.fr (F.C.); maeva.giraudo@obs-banyuls.fr (M.G.); alice.rodrigues@obs-banyuls.fr (A.M.S.R.); philippe.lebaron@obs-banyuls.fr (P.L.); 2Sorbonne Université, CNRS, Fédération de Recherche, FR3724, Observatoire Océanologique, 66650 Banyuls-sur-Mer, France

**Keywords:** marine ecotoxicity, environmental fate, personal care products, *Pocillopora damicornis*, avobenzone, Symbiodiniaceae

## Abstract

The intensive use of sunscreen products has raised concerns regarding their environmental toxicity and the adverse impacts of ultraviolet (UV) filters on ecologically important coral communities. Prior metabolomic analyses on symbiotic coral *Pocillopora damicornis* exposed to the UV filter butyl methoxydibenzoylmethane (BM, avobenzone) revealed unidentified ions in the holobiont metabolome. In the present study, follow-up differential metabolomic analyses in BM-exposed *P. damicornis* detected 57 ions with significantly different relative concentrations in exposed corals. The results showed an accumulation of 17 BM derivatives produced through BM reduction and esterification. The major derivative identified C16:0-dihydroBM, which was synthesized and used as a standard to quantify BM derivatives in coral extracts. The results indicated that relative amounts of BM derivatives made up to 95% of the total BM (*w*/*w*) absorbed in coral tissue after 7 days of exposure. Among the remaining metabolites annotated, seven compounds significantly affected by BM exposure could be attributed to the coral dinoflagellate symbiont, indicating that BM exposure might impair the photosynthetic capacity of the holobiont. The present results suggest that the potential role of BM in coral bleaching in anthropogenic areas should be investigated and that BM derivatives should be considered in future assessments on the fate and effects of BM in the environment.

## 1. Introduction

The use of sunscreen products is a public health necessity. However, the ultraviolet (UV) filters contained in these products can enter the marine environment due to their direct release into coastal marine areas through recreational activities (e.g., swimming, snorkeling) and via wastewater effluents [1,2,3,4]. Some organic UV filters have been shown to be harmful to marine organisms, including coral species [5,6,7,8,9,10,11,12,13]. For example, Downs et al. (2016) reported bleaching in *Stylophora pistillata* larvae after 24-h of exposure to 2.28 μg/L benzophenone-3. Larval deformities were visible at 22.8 µg/L and above. These data triggered local bans on the sale of certain UV filters, including BP-3 and octinoxate, in Hawaii, in the Republic of Palau, and in several other coastal communities where coral reefs were experiencing death events [14,15]. Meanwhile, the FDA has designated 12 UV filters currently marketed in the United States as requiring further studies (GRASE Category III) on their safety as topical agents (Federal Register 84FR6204-6275, 2019-03019).

In this context, it is crucial to investigate the environmental toxicity of organic UV filters and better understand their effects on coral to help develop safer, more effective, and environmentally friendly sunscreen products in the future. In prior work, the relative toxicity of 10 UV filters was quantified in the reef-building coral *Pocillopora damicornis* exposed to 5 to 1000 µg/L of each compound during 7-d. The results showed that it was possible to quantify the toxicity of these filters on the animal by measuring the relative concentration of inflammation markers by metabolomic profiling [12]. Among the compounds tested, the UV filter butyl methoxydibenzoylmethane (BM, avobenzone; Figure 1) did not trigger coral inflammation at 1 mg/L. However, coral exposed to BM was subject to a metabolomic shift in which the relative concentration of undetermined ions increased in the global coral metabolome.

The goal of this present study was to further explore the metabolism of *P. damicornis* exposed to increasing concentrations of BM in order to accurately characterize and better understand the effects of BM on the coral holobiont metabolome.

## 2. Material and Methods

### 2.1. Pocillopora damicornis

Samples of coral *P. damicornis* were collected in Oman in 2014 (CITES permit 37/2014). Coral colonies were acclimated for more than one year in tanks at the Banyuls Oceanological Observatory (Banyuls-sur-Mer, France). Organisms were maintained in artificial seawater (ASW) prepared with reverse osmosis purified water and pharmaceutical grade Tropic Marin PRO REEF Salt (Wartenberg, Germany). Salinity was adjusted to 36 g/L, pH = 8, and the temperature was set at 24 °C, with a 10:14 h light: dark photoperiod (Ecotech Marine Radion G5 lamps-model XR15 Pro, Bethlehem, PA, USA), including a gradual rise and fall of one hour each. The light program was the Coral lab AB+ program with a lunar cycle. All experiments were conducted using the same culture conditions.

### 2.2. Coral Exposure, Extraction, Metabolomic Profiling, and Analyses

The methods of exposure, extraction, LC-ESI^+^-MS profiling, and differential metabolomic analysis have been detailed previously [10,12]. Briefly, *P. damicornis* nubbins (N = 5 per concentration) were exposed for 7-d to 5, 50, 300, and 1000 µg/L of BM diluted in DMSO (0.25% *v*/*v*). The experiment at 1000 µg/L was repeated to confirm the results obtained. Artificial seawater with the same concentration of DMSO was used as a control (N = 5). This concentration of DMSO did not have visible impacts on coral health after 7-d of exposure, and no significant difference was found in the metabolomic profiles in corals from the control with and without DMSO (data not shown).

Metabolomic profiles were compared between the exposed and control corals as detailed previously [10,12,13]; only metabolites that varied significantly in concentration to both BM exposure experiments were retained in this study. Annotations were performed manually as described in the results and discussion section and with the help of Sirius software when needed [16].

### 2.3. Standard C16:0-Dihydrobm (***7***)

Compound **7** was synthesized upon request by R&H Discovery (Grabels, France). Standard solution S1 was obtained by diluting a mixture of C16:0-dihydroBM (2.1 mg) and BM (2.8 mg) in acetonitrile (10 mL). Standard solution S2 was obtained by diluting S1 (0.5 mL) with acetonitrile to reach a total volume of 10 mL. Standard solutions S3, S4, and S5 were obtained by serial 10× dilutions of S2. The standard solutions S1-5 and a replicate of the coral extract exposed to 1000 µg/L of BM were analyzed together in the same conditions as coral LC-MS profiling.

The MS^2^ spectra and retention time of standard C16:0-dihydroBM and the compound annotated as such with coral exposed to BM were identical. Extracted ion chromatograms of [C16:0-dihydroBM + Na]^+^ and [BM + H]^+^ main tautomer provided relative integration values for both ions in S1-5 and, therefore, a relative response coefficient in ESI^+^-MS. When reporting the main ion peak area ([BM + H]^+^ and [C16:0-dihydroBM + Na]^+^) in the function of the mass concentration in the linear region (S1–3), the correlation line slope was approximately 3 times lower for BM (Peak Area = 3.81 × 10^8^ Concentration, R^2^ = 0.9999) than for C16:0-dihydroBM (Peak Area = 1.108 × 10^9^ Concentration, R^2^ = 0.9968) (Appendix A).

### 2.4. Predicted Toxicity of BM Derivatives Compared to BM

The prediction of BM derivatives toxicity in aquatic organisms was carried out using the ecological structure–activity relationships (ECOSAR V 2.2, U.S. Environmental Protection Agency, Washington, DC, USA) predictive model distributed by the EPA (https://www.epa.gov/tsca-screening-tools/ecological-structure-activity-relationships-ecosar-predictive-model (accessed on 29 March 2023)).

## 3. Results and Discussion

Differential metabolomic analysis in the coral *P. damicornis* identified 57 ions from which the relative concentration was significantly affected (*p* < 0.05) after 7-d of exposure to increasing doses of BM. Manual annotation was able to match these 57 ions to 30 compounds, as detailed below.

### 3.1. Annotation of BM Derivatives

The annotation results showed that 17 out of the 30 compounds identified in the exposed corals were BM derivatives (Figure 2 and Table 1). It could be established that compound **1** was BM itself, and it was possible to propose a structure for the majority of its derivatives.

The manual annotation procedure was similar for all the dihydroBM fatty acid esters (**2–11**) and is detailed hereafter for compound **7** as an example. ESI^+^-MS and the fragmentation spectra for all compounds can be found in the Appendix A. Compound **7** was the major compound among the dihydroBM fatty acid esters. First, the ion at *m*/*z* 573.3916 was determined to be a sodium adduct, the formula of which was C_36_H_54_O_4_Na^+^ (Calcd. 573.3914), corresponding to a compound with 10 degrees of unsaturation. In the MS^2^ spectrum of the sodium adduct, the first neutral loss corresponded to a loss by the β-elimination of palmitic acid (C_16_H_34_O_2_, C16:0) and led to the product at *m*/*z* 317.1513, the formula for which was C_20_H_22_O_2_Na^+^ (Calcd. *m*/*z* 317.1512) (Figure 3). This cation corresponded to a sodiated deoxy-BM. These two elements suggested that one of the two ketones of BM had been reduced in vivo and then esterified with palmitic acid naturally present in the coral to ultimately form a dihydroBM ester. This hypothesis was supported by the observation of a product at *m*/*z* 135.0441 (C_8_H_7_O_2_^+^, calcd. *m*/*z* 135.0440) corresponding to the (4-methoxybenzylidyne)oxonium acylium cation indicating that this BM subunit had not been modified. Similarly, the (4-(*tert*-butyl)benzylidyne)oxonium acylium cation was present in the fragmentation spectrum of compound **8** (Appendix A). Altogether, these data indicated that BM was modified at the 1,3-diketo subunit. Compound **7** was obtained by synthesis and compared to the product annotated in BM-exposed corals; the results showed that both compounds were strictly identical (Appendix A).

Analogs **2–11** corresponded to compounds in which BM was reduced and then acylated with various fatty acids. Each time, the major isomer was the product resulting from the reduction and subsequent acylation of ketone 3 (Figure 1), which is consistent with this ketone being more electrophilic. However, the reduction in ketone 1 was observed systematically. It produced the minor isomers of each of the compounds listed in Table 1.

Isomeric structures were proposed for compounds **12** and **13** based on their mass spectrometry data (Appendix A). These two compounds were very close structurally and present exclusively in BM-exposed coral. The protonated molecular ion of compound **12** was observed at *m*/*z* 633.3218, as well as the corresponding sodium, potassium and dimethylammonium adducts, respectively, at *m*/*z* 655.3034, 671.2771, and 678.3794, (Figure 4). Two fragments were detected in ESI^+^-MS at *m*/*z* 311.1643 (Frag. 1, corresponding to the protonated molecular ion of BM, C_20_H_23_O_3_^+^) and 323.1643 (Frag. 2, C_21_H_23_O_3_^+^), both resulting from the in-source fragmentation of the protonated molecular ion of compound **12**. Moreover, the molecular ion at *m*/*z* 633.3218 with the formula C_41_H_45_O_6_^+^ (calcd. 655.3030) corresponded exactly to the sum of the formulas of the two product ions Frag. 1 and 2. The MS^2^ spectra of Frag. 1 and Frag. 2 provided further information allowing the annotation of compounds **12** and **13** (Appendix A).

The collision-induced fragmentation of Frag. 1 was identical to that of the BM protonated molecular ion (Appendix A) and Frag. 1 was, therefore, annotated as [BM + H]^+^. The collision-induced fragmentation of Frag. 2 produced both (4-methoxybenzylidyne)oxonium and (4-(*tert*-butyl)benzylidyne)oxonium ions at *m*/*z* 135.0443 and 161.0963, respectively (Appendix A). This fragment was thus annotated as “[BM + C + H]^+”^ in which the extra carbon was located on the central part. The carbon addition on BM could result from the formation of an aldol condensation product between BM and formaldehyde. This highly electrophilic product could have produced two isomers **12** and **13** through the 1,4-nucleophilic addition of a second molecule of BM (Frag. 1). Compounds **15**–**17** could not be annotated.

A joint analysis of synthetic compound **7** with BM established that, in equal amounts, [**7** + Na]^+^ ion peak integration was one to four times higher than that of [BM + H]^+^ (Appendix A). At higher concentrations, the relative peak integration of [**7** + Na]^+^ decreased, but at concentrations where the detector response was linear (<1.5 µg/L), the coefficient of linear correlation ratios in the graphs represented peak areas versus concentration and was 2.9 in favor of [**7** + Na]^+^ (Appendix A). Thus, it could be estimated that the relative integrations of compounds **2**–**11** as sodium adducts should be three times greater than that of [BM + H]^+^ for the same mass concentration. On this basis, the relative amounts of the various dihydroBM esters to BM could be estimated (Figure 5, Appendix A).

The results showed a dose-dependent accumulation of BM derivatives in coral at the end of 7-d exposure, with 1.3 to 32 times higher levels of dihydroBM esters relative to BM. These observations indicate that BM can accumulate in corals mostly as derivatives. One mechanism by which xenobiotics are converted into less toxic, excretable forms in the body of exposed organisms involves their conjugation with endogenous polar metabolites. Conjugation with lipids seems undesirable as it generates more lipophilic conjugates that accumulate in tissues and may exhibit higher toxicity [17]. The toxicity of dihydroBM ester derivatives predicted using the ecological structure–activity relationships (ECOSAR) model indicated acute (short-term) toxicity, 1000 to 900,000 times greater in aquatic organisms than for the parent compound (details in supporting information). Similar observations were made in a microalga exposed to the flame-retardant triphenyl phosphate (TPHP), where the biotransformation product generated by palmitoyl conjugation showed increased predicted toxicity compared with the parent compound [18]. Further analysis is needed to verify whether the transformation process of BM is specific to corals. Nevertheless, the possible occurrence and toxicity of dihydroBM esters should be considered when measuring the bioaccumulation of BM in marine organisms.

### 3.2. Annotation of Coral Metabolites

The relative concentration of 13 coral metabolites significantly increased or decreased after 7-d of exposure to BM, with the lowest observed effect concentration at 300 µg/L. These metabolites are listed in Table 2 and are grouped by compound class. The corresponding MS^2^ spectra can be found in the Appendix A.

The exposure of corals to 300 and 1000 µg/L of BM significantly increased the concentration of three ceramides, which were annotated as follows (only compound **18** is detailed as an example). The protonated molecular ion [**18** + H]^+^ was found at *m*/*z* 548.4674. The formula corresponding to this mass was C_34_H_62_NO_4_^+^ (Calcd. 548.4673). It was possible to observe in the MS^2^ spectrum of this ion both the product from the C16 fatty acid chain (C_16_H_30_NO^+^ at *m*/*z* 252.2320) and those resulting from the fragmentation of the C18 sphingoid base (C_18_H_27_^+^ at *m*/*z* 243.2112, C_18_H_30_N^+^ at 260.2373, C_18_H_30_NO^+^ at 278.2473, C_18_H_32_NO_2_^+^ at 296.2575 and C_18_H_34_NO_3_^+^ at 312.2561) (Appendix A). The fragmentation pattern of the sphingoid base suggested that it must be trihydroxylated, and therefore, the C16 fatty acid was not hydroxylated but doubly unsaturated (C16:2). However, uncertainties remained for the other ceramides, and we chose not to specify the locations of the hydroxyls and double bonds in the annotations.

Ceramides are major components of cell membranes and are ubiquitous in eukaryotic organisms. Ceramides with a C19 sphingoid base are rare, but an identical or isomeric compound of compound **19** has been previously described as a fungus of the genus *Cordyceps* [19]. Analogs with additional hydroxyls on the sphingoid base were described in the same study, as well as analogs where the sphingoid base was deoxygenated in position 1, highlighting the fact that the position of the hydroxyl groups could not be annotated more precisely based on the literature precedents.

Although ceramides cannot be assigned to either holobiont partner, an increased relative concentration of three ceramides in BM-exposed corals may be a sign of poor holobiont health. Ceramides are involved in cellular communication and the coordination of the cellular response to external stimuli [20]. In vertebrates, ceramide levels are elevated in response to diverse cellular stress, and they play a central role in apoptosis [21]. There is very limited information on the role of ceramides in corals and in invertebrates in general. A study of the Pacific oyster (*Crassostrea gigas*) suggested that ceramides may have analogous functions in vertebrates and invertebrates [22]. Further studies are needed in corals to better understand the role of ceramides in response to BM exposure.

Compounds **21** and **22** could not be annotated precisely. Their molecular formulas were C_50_H_79_NO_11_ for **21** (Calcd. *m*/*z* for C_50_H_80_NO_11_^+^ 870.5726, 12 degrees of unsaturation) and C_44_H_80_NO_11_^+^ for **22** (Calcd. *m*/*z* for C_44_H_80_NO_11_^+^ 798.5726, 6 degrees of unsaturation). The two molecular ions of these metabolites fragmented extensively in collision-induced MS^2^ spectroscopy, precluding formal annotation. However, three characteristic ions at *m*/*z* 60.0808, 86.0964, and 104.1070 (100%) corresponding to the cations HNMe_3_^+^, Me_3_N^+,^-CH=CH_2,_ and Me_3_N^+^-CH-CH_2_OH, respectively, suggested that they may be choline derivatives. The relative concentration of metabolites **21** and **22** increased significantly at 300 µg/L BM and above.

Compounds **23**–**27** could be annotated as monogalactosyldiacylglycerols (MGDGs) as previously described in [23]. Briefly, the fragment resulting from *sn*-1 fatty acid loss exhibited a higher peak intensity than the one resulting from *sn*-2 loss from the sodiated adduct of MGDGs. The relative concentration of four of these MGDGs increased significantly as a result of the exposure of coral to BM from 300 µg/L, whereas the relative concentration of compound **25** decreased significantly (Table 2). MGDGs, DGDGs (digalactosyldiacylglycerols), and SQDGs (sulfoquinovosyldiacylglycerols) were the precursors for chloroplast lipids that formed symbiotic dinoflagellates thylakoid membranes [24]. Although only MGDGs were detected in the differential analyses of the exposed and control corals, their relative concentration changes suggested chloroplast degradation.

The concentration of two carotenoids (compounds **28** and **29**) decreased significantly in response to 300 and 1000 µg/L of BM (Table 2). The molecular formulas of isomeric compounds **28** and **29** were C_39_H_48_O_6_ (Calcd. *m*/*z* for C_39_H_49_O_6_^+^ 613.3524, 16 degrees of unsaturation). There are very few known natural products with this molecular formula. The most common is pyrrhoxanthin: a peridinin analog. Sirius annotated this compound as pyrrhoxanthin (60% CSI:FingerID score) [16,25], but the experimental MS^2^ spectrum of the protonated molecular ion of pyrrhoxanthin is not available in public databases. Pyrrhoxanthin is relatively specific to Dinophyceae microalgae [26,27,28,29], whereas the isomers known to date have been isolated from clams of the genus *Corbicula* [30]. It is, therefore, reasonable to consider that one of the two isomers, **28** and **29**, should be pyrrhoxanthin, the other being a new compound or a compound already characterized in *Corbicula* sp. Carotenoids are required for the formation of photosynthetic pigment–protein complexes of the thylakoid membrane. The decrease in the relative concentration of these two carotenoids might be indicative of an alteration in the photosynthetic capacity of the coral Symbiodiniaceae.

Overall, these results corroborated those obtained with planktonic microalga [31]. *Tetraselmis* sp. cells cultured in the presence of BM experienced an increase in cell volume and were granularity detected by cytometry. BM also induced a significant decrease in chlorophyll, a cell fluorescence in this species, indicating that the photosynthetic capacity was impacted while the growth rate (number of cells) was not. Thus, there is a body of evidence that exposure to BM may have altered the photosynthetic capacity of the Symbiodiniaceae and, in turn, the holobiont.

## 4. Conclusions

Exposure of the coral *P. damicornis* to BM induced a non-typical metabolome alteration relative to what was observed for octocrylene or ethylhexyl salicylate [10,12]. A close examination of the metabolites that varied significantly in concentration compared to the healthy control provided insight into the origin of the metabolic differences. On the one hand, we were able to demonstrate that coral accumulated BM derivatives resulting from the reduction followed by the in vivo esterification of BM. These compounds were present in significant amounts relative to BM itself and showed increased predicted toxicity in aquatic organisms compared to the parent compound, warranting their consideration for the environmental monitoring of BM’s presence, fate, and effects. On the other hand, this work has also demonstrated that the photosynthetic capacity of coral Symbiodiniaceae might be altered by the exposure of the coral to this xenobiotic. This effect was detected at the somewhat high BM concentration of 300 µg/L. It would be important to measure the levels of BM contamination in the reefs and surrounding waters of anthropized areas in order to evaluate whether BM could contribute locally to coral bleaching.

## Figures and Tables

**Figure 1 metabolites-13-00533-f001:**
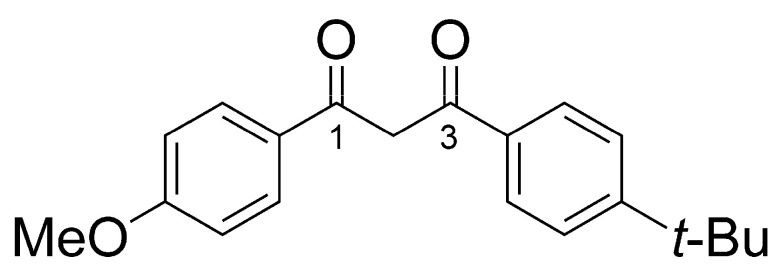
Chemical structure of BM (diketo tautomer).

**Figure 2 metabolites-13-00533-f002:**
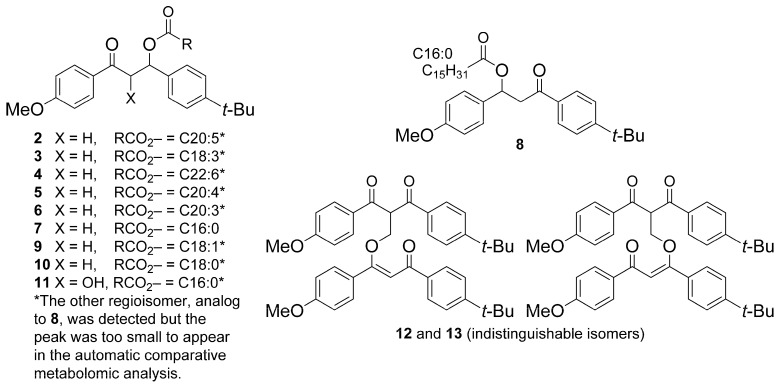
Proposed structures for compounds **2**–**13** identified in *P. damicornis* metabolome after 7-d exposure to 5–1000 µg/L of BM. The structure of compound **7** was firmly confirmed by comparison with a synthetic standard.

**Figure 3 metabolites-13-00533-f003:**
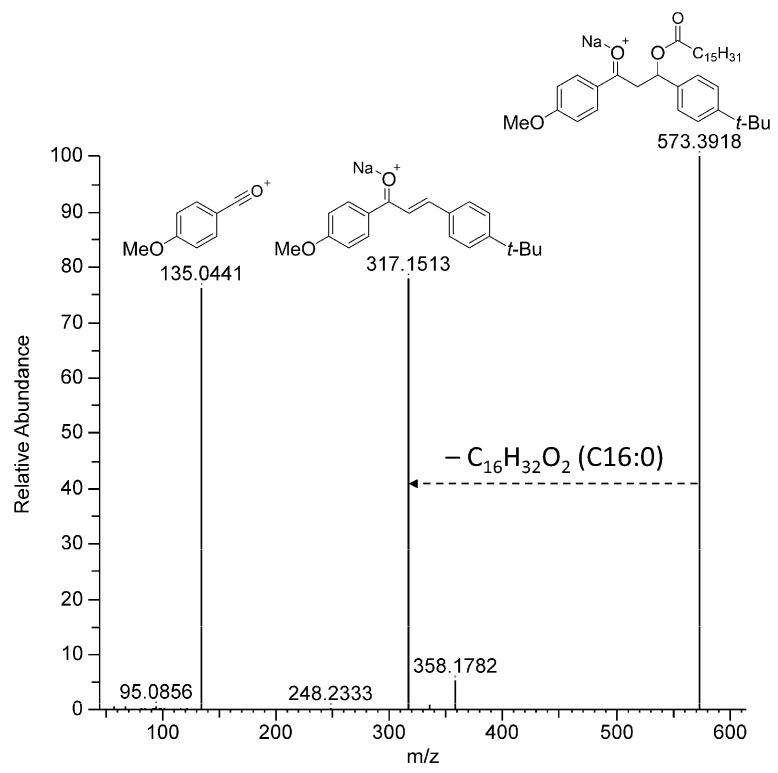
Collision-induced dissociation of compound **7** sodium adduct.

**Figure 4 metabolites-13-00533-f004:**
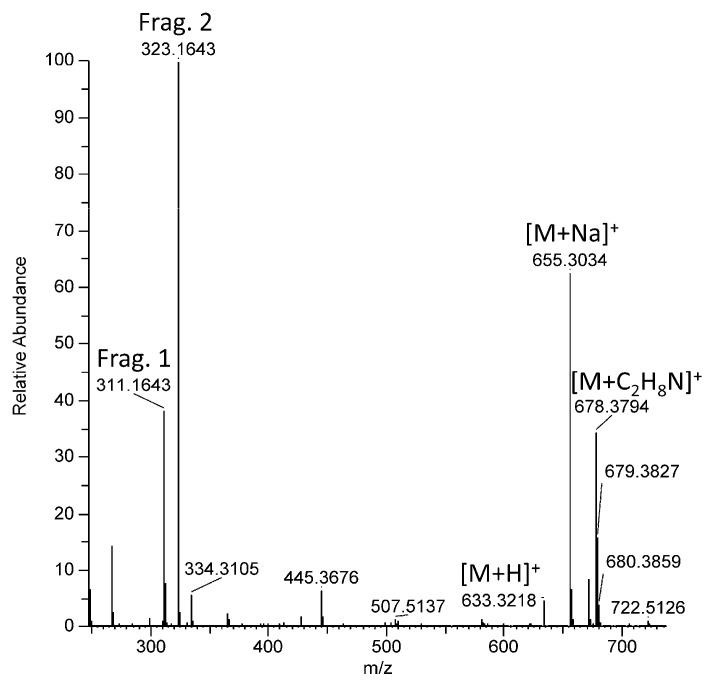
Representative part of the ESI^+^-MS spectrum of **12** (from Appendix A) showing the main adducts and the two fragments detected.

**Figure 5 metabolites-13-00533-f005:**
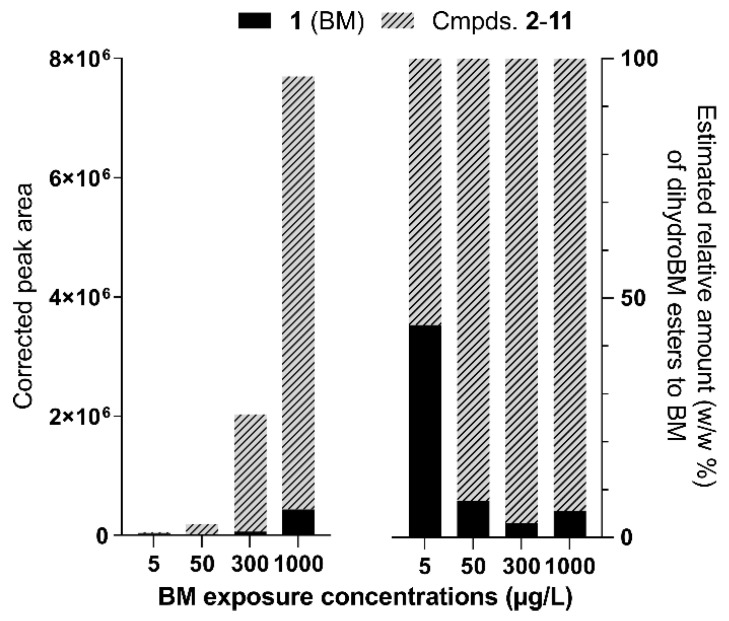
Sum of compounds **1**–**11** extracted ion chromatogram peak areas. DihydroBM esters peak areas were corrected (divided by 3) to account for the relative ESI^+^-MS response of protonated BM (**1**) and dihydroBM esters sodium adducts (**2**–**11**) (**left**); estimated relative amount (*w*/*w*%) of BM and dihydroBM esters in coral tissues (**right**).

**Table 1 metabolites-13-00533-t001:** List of BM derivatives detected in *P. damicornis* after 7-d exposure to 5–1000 µg/L of BM.

Cmpd. #	t_R_ (Min)	Exp. *m*/*z*	Ion Species	Th. *m*/*z*	Molecular Formula
**1** (BM)	7.62/9.32	311.1642	[M + H]^+^	311.1642	C_20_H_22_O_3_
**2**	13.18	619.3762	[M + Na]^+^	619.3758	C_40_H_52_O_4_
**3**	13.49	595.3758	[M + Na]^+^	595.3758	C_38_H_52_O_4_
**4**	13.55	645.3916	[M + Na]^+^	645.3914	C_42_H_54_O_4_
**5**	13.86	621.3916	[M + Na]^+^	621.3914	C_40_H_54_O_4_
**6**	14.42	623.4072	[M + Na]^+^	623.4071	C_40_H_56_O_4_
**7** ^ a^	14.87	573.3916	[M + Na]^+^	573.3914	C_36_H_54_O_4_
**8**	14.94	573.3914	[M + Na]^+^	573.3914	C_36_H_54_O_4_
**9**	14.98	599.4071	[M + Na]^+^	599.4071	C_38_H_56_O_4_
**10**	15.91	601.4229	[M + Na]^+^	601.4227	C_38_H_58_O_4_
**11**	12.77	589.3864	[M + Na]^+^	589.3863	C_36_H_54_O_5_
**12**	10.39	655.3033	[M + Na]^+^	655.3030	C_41_H_44_O_6_
**13**	10.47	655.3034	[M + Na]^+^	655.3030	C_41_H_44_O_6_
**14** ^ b,c^	9.32	449.1370	n.d.	n.d.	n.d.
**15** ^ c^	13.69	283.1694	[M + H]^+^	283.1693	C_19_H_22_O_2_
**16** ^ c^	14.35	559.3760	[M + Na]^+^	559.3758	C_35_H_52_O_4_
**17** ^ c^	14.45	279.1744	[M + H]^+^	279.1743	C_20_H_22_O

^a^ Identical to a synthetic standard. ^b^ n.d.: not determined. ^c^ These compounds could not be annotated.

**Table 2 metabolites-13-00533-t002:** Metabolites with significantly different relative concentrations in *P. damicornis* exposed to BM for 7-d, organized by compound class.

Cmpd. #	t_R_ (Min)	Exp. *m*/*z*	Species	Th. *m*/*z*	Molecular Formula	Cmpd Class ^a^	Annotation ^b^	Rel. Conc. (BM Conc.) ^c^
**18**	12.61	548.4674	[M + H]^+^	548.4673	C_34_H_61_NO_4_	Ceramide	Cer(18/16) derivative	↗ (300)
**19**	13.08	562.4830	[M + H]^+^	562.4830	C_35_H_63_NO_4_	Ceramide	Cer(19/16) derivative	↗ (300)
**20**	13.38	550.4832	[M + H]^+^	550.4830	C_34_H_63_NO_4_	Ceramide	Cer(18/16) derivative	↗ (1000)
**21**	8.45	870.5731	[M + H]^+^	870.5726	C_50_H_79_NO_11_	Choline deriv.	n.d.	↗ (300)
**22**	9.35	798.5731	[M + H]^+^	798.5726	C_44_H_79_NO_11_	Choline deriv.	n.d.	↗ (300)
**23**	10.93	789.4553	[M + Na]^+^	789.4548	C_45_H_66_O_10_	MGDG	MGDG 18:5/18:5	↗ (1000)
**24**	11.23	791.4709	[M + Na]^+^	791.4705	C_45_H_68_O_10_	MGDG	MGDG 18:5/18:4	↗ (300)
**25**	11.66	817.4853	[M + Na]^+^	817.4861	C_47_H_70_O_10_	MGDG	MGDG 18:5/20:5	↘ (300)
**26**	12.31	769.4871	[M + Na]^+^	769.4861	C_43_H_70_O_10_	MGDG	MGDG 18:5/16:1	↗ (300)
**27**	12.67	771.5025	[M + Na]^+^	771.5018	C_43_H_72_O_10_	MGDG	MGDG 18:4/16:1	↗ (300)
**28**	11.13	613.3521	[M + H]^+^	613.3524	C_39_H_48_O_6_	Carotenoid	Pyrrhoxanthin or isomer	↘ (300)
**29**	11.30	613.3525	[M + H]^+^	613.3524	C_39_H_48_O_6_	Carotenoid	Pyrrhoxanthin or isomer	↘ (1000)
**30**	12.23	1070.6836	[M + NH_4_]^+^	1070.6833	C_53_H_96_O_20_	n.d.	n.d.	↗ (300)

^a^ MGDG: Monogalactosyldiacylglycerol; n.d.: not determined. ^b^ For ceramides, sphingoid base and fatty acid chain lengths were visible in the MS^2^ spectra, but the position of additional hydroxyl groups and double bonds remained speculative and are, therefore, deliberately not specified in the annotations. ^c^ The relative concentration increased (↗) or decreased (↘) significantly (*p* < 0.05) at the BM concentration noted in parenthesis and above (in µg/L).

## Data Availability

Full data for compounds annotation are provided in the supporting information. Raw data are available upon request to the corresponding author.

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
