# Peer review of "On the Fate of Butyl Methoxydibenzoylmethane (Avobenzone) in Coral Tissue and Its Effect on Coral Metabolome"

_metabolites, 2023, doi:10.3390/metabo13040533_

Round 1

Reviewer 1 Report

Reef-building corals serve as a foundation for ecosystems in tropical and subtropical coastal regions. However, anthropogenic activities and global climate changes pose significant threats to coral ecosystems. Recently, excessive use of UV sunscreen by tourists has been considered toxic to reef-building corals, potentially leading to coral bleaching - a physiological symptom of coral decline. Commercial sunscreen products often contain UV-absorbing chemical compounds, such as oxybenzone and octinoxate. In this study, the authors explored the coral metabolism of the UV-absorbing compound avobenzone (BM, butyl methoxydibenzoylmethane) in the symbiotic coral Pocillopora damicornis. After exposure to BM, the authors identified 57 ions and 17 BM derivatives.

Before the publication of this paper, I would request that the authors consider the following points. While the results presented in the paper are informative regarding the physiological impacts of BM on reef-building corals, I would like to point out that the physiological arguments made are mostly overspeculations and cannot be fully judged based solely on the presented results. For example, in the paper's conclusion, the authors state that "this work has also demonstrated that the photosynthetic capacity of coral Symbiodiniaceae was altered by exposure of the coral to this xenobiotic" (L269-270), although there is no photosynthetic capacity data presented in the manuscript nor any description in Materials and Methods. While the results of the analytical study using MS spectrometry are convincing, overspeculative statements on physiological issues could detract from the paper's overall quality. Therefore, I advise the authors to remove the physiological statements from the manuscript.

If the authors wish to discuss physiological issues, including possible linkage with coral bleaching, I request that they consider the following issues:

(1) When avobenzone absorbs UV light, it transitions to an excited state and often generates reactive oxygen species (ROS). ROS can potentially cause cell damage and inflammation. Thus, it is necessary to distinguish indirect toxicity, such as ROS production, from direct metabolic impacts of BM and its derived chemical species (e.g., interpretation of the decrease in the compounds 28 and 29 in Table 2).

(2) Avobenzone is known to naturally photodegrade. Photodegradation involves the absorption of light, which produces degradation products. These degradation products are more stable and safer because they have lower photosensitivity and toxicity. Therefore, it is essential to discuss the derived chemical species in reference to the BM photodegradation intermediates and products reported.

(3) The water insolubility of BM may make it virtually impossible to access underwater corals. It might be possible to think that corals might accumulate BM through phytoplankton taken up BM flowing on the surface of water. Similarly, it is crucial to differentiate between the host coral and its symbiotic algae metabolisms.

Thus, I would like to request that the authors should focus on the analytical study and minimize overspeculative statements on physiological issues. If the authors wish to discuss physiological issues, they should consider the above-mentioned points and provide a more detailed analysis.

Author Response

Reviewer 1:

Reef-building corals serve as a foundation for ecosystems in tropical and subtropical coastal regions. However, anthropogenic activities and global climate changes pose significant threats to coral ecosystems. Recently, excessive use of UV sunscreen by tourists has been considered toxic to reef-building corals, potentially leading to coral bleaching - a physiological symptom of coral decline. Commercial sunscreen products often contain UV-absorbing chemical compounds, such as oxybenzone and octinoxate. In this study, the authors explored the coral metabolism of the UV-absorbing compound avobenzone (BM, butyl methoxydibenzoylmethane) in the symbiotic coral Pocillopora damicornis. After exposure to BM, the authors identified 57 ions and 17 BM derivatives.

Before the publication of this paper, I would request that the authors consider the following points. While the results presented in the paper are informative regarding the physiological impacts of BM on reef-building corals, I would like to point out that the physiological arguments made are mostly overspeculations and cannot be fully judged based solely on the presented results. For example, in the paper's conclusion, the authors state that "this work has also demonstrated that the photosynthetic capacity of coral Symbiodiniaceae was altered by exposure of the coral to this xenobiotic" (L269-270), although there is no photosynthetic capacity data presented in the manuscript nor any description in Materials and Methods. While the results of the analytical study using MS spectrometry are convincing, overspeculative statements on physiological issues could detract from the paper's overall quality. Therefore, I advise the authors to remove the physiological statements from the manuscript.

We would like to say that metabolomics-based diagnostic is more and more commonly used in Human pathologies (we have a paper currently under evaluation on this topic) and that metabolomics signatures are more sensitive than many observations of physiological endpoints. It is our job to make metabolomics a more widely accepted method for NOEC evaluation in ecotoxicology as many colleagues do not understand it yet (see comments of industry on our previous articles). Metabolomics is great for NOEC measurement because it detects signals when no physiological endpoint is visible yet. However, we do agree that our physiological statements should be more moderate. As an example, when we see that the concentration of chlorophyll derivatives decreases, it means that the exposition of the holobiont to BM has an impact, meaning that BM NOEC is below the tested concentration. This is the key information. Then the question as to what exactly is the effect may not be resolved (at it is not our point here), but we all know that one new finding opens new questions and it is not inappropriate in a scientific paper to raise concern about a possible impact on the photosynthetic capacity of the holobiont. So, your critic is appropriate, but if you don’t mind we would like to maintain our comments written rather as speculations than as affirmations.

The following sentences have been modified accordingly to your comment:

  • Among the remaining metabolites annotated, seven compounds significantly affected by BM exposure could be attributed to the coral dinoflagellate symbiont, indicating an impact on the photosynthetic capacity of the animal. à Among the remaining metabolites annotated, seven compounds significantly affected by BM exposure could be attributed to the coral dinoflagellate symbiont, indicating that BM exposure might impair the photosynthetic capacity of the holobiont.
  • The present results suggest that BM may play a role in coral bleaching in anthropized areas and that BM derivatives should be included in future evaluations of BM environmental fate and effects. à The present results suggest that the potential role of BM in coral bleaching in anthropogenic areas should be investigated, and that BM derivatives should be considered in future assessments of the fate and effects of BM in the environment.
  • The decrease in the relative concentration of these two carotenoids could be indicative of an alteration in the photosynthetic capacity of the coral Symbiodiniaceae. à The decrease in the relative concentration of these two carotenoids might be indicative of an alteration in the photosynthetic capacity of the coral Symbiodiniaceae.
  • On the other hand, this work has also demonstrated that the photosynthetic capacity of coral Symbiodiniaceae was altered by exposure of the coral to this xenobiotic. à On the other hand, this work has also demonstrated that the photosynthetic capacity of coral Symbiodiniaceae might be altered by exposure of the coral to this xenobiotic.

If the authors wish to discuss physiological issues, including possible linkage with coral bleaching, I request that they consider the following issues:

(1) When avobenzone absorbs UV light, it transitions to an excited state and often generates reactive oxygen species (ROS). ROS can potentially cause cell damage and inflammation. Thus, it is necessary to distinguish indirect toxicity, such as ROS production, from direct metabolic impacts of BM and its derived chemical species (e.g., interpretation of the decrease in the compounds 28 and 29 in Table 2).

(2) Avobenzone is known to naturally photodegrade. Photodegradation involves the absorption of light, which produces degradation products. These degradation products are more stable and safer because they have lower photosensitivity and toxicity. Therefore, it is essential to discuss the derived chemical species in reference to the BM photodegradation intermediates and products reported.

Regarding your previous comment and your first two points here, these are not really the questions we are addressing with these experiments. However, we would like to highlight the last sentence before conclusion: Overall, these results corroborate those obtained with the planktonic microalga Tetraselmis sp. for which exposure to BM increased granularity and cell volume while decreasing cell fluorescence [29].

In fact, we do have physiological endpoints on one microalga. Although the species is different, the conclusions are identical if we compare metabolomics with more widely accepted cytometry, and we think it is important to point this out. Maybe this statement as a last sentence was not clear enough to the reader and we propose to change it into a specific paragraph as follows:

Overall, these results corroborate those obtained with a planktonic microalga [29]. Tetraselmis sp. cells cultured in the presence of BM have experienced an increase in cell volume and granularity detected by cytometry. BM also induced a significant decrease of chlorophyll a cell fluorescence in this species, indicating that the photosynthetic capacity was impacted while the growth rate (number of cells) was not. Thus, there is a body of evidence that exposure to BM may alter the photosynthetic capacity of the Symbiodiniaceae, and the holobiont in turn.

 (3) The water insolubility of BM may make it virtually impossible to access underwater corals. It might be possible to think that corals might accumulate BM through phytoplankton taken up BM flowing on the surface of water. Similarly, it is crucial to differentiate between the host coral and its symbiotic algae metabolisms.

There is plenty of evidence that underwater corals can accumulate compounds that are much more lipophilic that BM. Octocrylene, for example, has been found in coral in many places around the world. FYI, we have measured and published the solubility of BM is sea water. It is 47 mg/L. This is not much, but it is way above the NOEC measured on coral in the present article.

Thus, I would like to request that the authors should focus on the analytical study and minimize overspeculative statements on physiological issues. If the authors wish to discuss physiological issues, they should consider the above-mentioned points and provide a more detailed analysis.

Reviewer 2 Report

A paper entitled “On the fate of butyl methoxydibenzoylmethane (avovenzone) in coral tissue and its effect on coral metabolome” is submitted to Metabolites for further reviewing and publication. In general, this manuscript is written very well. The authors described the effect of BM in coral tissue. It is very interesting in environment toxicity. I recommended that this manuscript is acceptable for publication with its present form.

Minor comment

1. Figure 2, in the structure of 11, the stereogenic center of compound 11 should be determined. Was this compound existed with enantiomers ?

Author Response

Reviewer 2:

A paper entitled “On the fate of butyl methoxydibenzoylmethane (avovenzone) in coral tissue and its effect on coral metabolome” is submitted to Metabolites for further reviewing and publication. In general, this manuscript is written very well. The authors described the effect of BM in coral tissue. It is very interesting in environment toxicity. I recommended that this manuscript is acceptable for publication with its present form.

Thank you for the nice feedback on our work.

Minor comment

  1. Figure 2, in the structure of 11, the stereogenic center of compound 11should be determined. Was this compound existed with enantiomers?

I suppose you mean diastereoisomers. It is difficult to be sure because all of these compounds will have one regioisomer (thus with same m/z) resulting from the reaction occurring on the left ketone too. For example, the extracted ion chromatogram for compound 7 gives two peaks, the major one is 7 and the minor one 8. Compound 11 has 4 possible isomers (regio- and diastereosimers) then, each on them maybe existing as a pair of enantiomers. We do see 4 peaks in LCMS, one large one (11), one medium and two smaller ones. Enantiomers could only be determined upon isolation or separation with chiral chromatography and comparison with standards (that do not exist). Diastereoisomers should also be isolated or compared with standards. Any isolation attempt would be very risky owing to the very small relative amount of this compound (not to say impossible) and would require to expose large colonies of coral and we would rather not do that. Chemical synthesis is also very complex for a questionable (according to us) benefit regarding the objectives of the work.

Author Response

Reviewer 3:

General comments This manuscript reports the accumulation and metabolic conversion of the sunscreen compound “avobenzone” (BM) in a coral species with reference to the possible degradation of chloroplasts of the coral’s photosynthetic symbiont. The exposure experiment of coral to BM was well-designed and conducted. The metabolomic analysis of the BM-exposed coral was well-designed and performed, too. The annotation of metabolites was also well done.

We appreciate the nice feedback.

However, disagreements in the text should be corrected. For example, while the authors could “match 38 compounds as detailed below” (L102-103 and L105), only 30 compounds were annotated/tried in Tables 1 and 2, the main text, and the supplementary material.

Sorry, this is a typing mistake that went thru all proof readings. Thank you for pointing this out!

More specific comments are shown below. The manuscript thus needs to be revised accordingly.

 Specific Comments L18-19 “Results indicated that relative amounts of BM made up to 95% of the total BM (w/w) in coral tissue after 7 days of exposure.” According to my interpretation, ~95% of the added BM was absorbed/adsorbed by coral.

There was a word (derivatives) missing in this sentence. That you for pointing this out. We changed it as follows to make it clearer: Results indicated that relative amounts of BM derivatives made up to 95% of the total BM (w/w) absorbed in coral…

Moreover, I would like to know if some amount of the BM was adsorbed to and retrieved from nonliving parts of the coral, for example, stony corallite.

This is impossible to say if you look at our protocol. What we know is that BM is taken up by holobiont biomachinery, so at some point it reaches coral tissues. Whether or not a portion of this BM (maybe the one that is not metabolized) is in fact absorbed on the corallite, we don’t know since we extract all at once (they are small pieces).

L240-241 “MGDGs are one of the major constituents of thylakoid of coral endosymbionts [22]” Ref [22] states, “In the zooxanthellae fractions, 21 molecular species of three GL classes (MGDG, DGDG, and SQDG) were identified.” In addition to MGDG, DGDG (digalactosyldiacylglycerol) and SQDG (sulfoquinovosyldiacylglycerol) are listed as the three major glycolipid components of the thylakoid membrane. However, DGDG and SQDG are not mentioned in this manuscript. The authors should refer to and discuss the presence/absence of DGDG and SQDG.

The text was changed according to your suggestion.

L265-266 “… coral accumulated BM derivatives resulting from the reduction followed by in vivo esterification of BM.” Discussing the biological function of BM esterification with palmitic acid and other fatty acids should be appropriate here.

This is an interesting suggestion. The text has been modified substantially to discuss upon the occurrence of these fatty acid conjugates, both at the end of the section describing BM-fatty acid conjugates and in the conclusion. Two references have been added too.